# Integrating Sparse Learning-Based Feature Detectors into Simultaneous Localization and Mapping—A Benchmark Study

**DOI:** 10.3390/s23042286

**Published:** 2023-02-18

**Authors:** Giuseppe Mollica, Marco Legittimo, Alberto Dionigi, Gabriele Costante, Paolo Valigi

**Affiliations:** 1Dipartimento di Ingegneria, Università degli Studi di Perugia, 06125 Perugia, Italy; 2ART S.p.A Company, 06065 Perugia, Italy

**Keywords:** learning-based features detectors, simultaneous localization and mapping (SLAM), vision-based pose estimation, deep learning

## Abstract

Simultaneous localization and mapping (SLAM) is one of the cornerstones of autonomous navigation systems in robotics and the automotive industry. Visual SLAM (V-SLAM), which relies on image features, such as keypoints and descriptors to estimate the pose transformation between consecutive frames, is a highly efficient and effective approach for gathering environmental information. With the rise of representation learning, feature detectors based on deep neural networks (DNNs) have emerged as an alternative to handcrafted solutions. This work examines the integration of sparse learned features into a state-of-the-art SLAM framework and benchmarks handcrafted and learning-based approaches by comparing the two methods through in-depth experiments. Specifically, we replace the ORB detector and BRIEF descriptor of the ORBSLAM3 pipeline with those provided by Superpoint, a DNN model that jointly computes keypoints and descriptors. Experiments on three publicly available datasets from different application domains were conducted to evaluate the pose estimation performance and resource usage of both solutions.

## 1. Introduction

Autonomous driving is certainly one of the most popular research directions in the robotics and intelligent transportation communities. The core capabilities of an autonomous driving agent are grounded on the navigation stack, which is composed of the following components: i.e., localization, mapping, path planning and collision avoidance. Among these, the localization module is probably the most crucial one, being the prerequisite for the proper functioning of all the others. Therefore, its performance is of utmost importance for the success of the overall navigation pipeline.

Localization aims to estimate and describe the agent pose, and i.e., its position and orientation in 3D space. This information is extracted from the input data provided by the available sensors of the agent, such as LIDAR, lasers, monocular or stereo cameras, and IMUs. Cameras are particularly attractive, due to their low cost, weight, and energy consumption, and the significant amount of information about the surrounding environment contained in the images collected. Stereo vision is probably the most common configuration, and several works and commercial products have proven its effectiveness. Nonetheless, its accuracy is tightly linked to the correctness of the stereo calibration, the estimated parameters of which might drift over time, due to physical modification of the rig (e.g., thermal expansion and cool contraction). The monocular setup considered in this paper does not suffer from this limitation. This advantage comes at the cost of more challenging image processing algorithms (He et al. [1]), due to the well-known scale drift and the single-point-of-failure (SPOF) problems (Aqel et al. [2], Yang et al. [3]).

From a methodological point of view, localization is achieved by relying on Visual Odometry (VO) or Simultaneous Localization And Mapping (SLAM), as presented in numerous literature works e.g., Yousif et al. [4], Agostinho et al. [5] and Chen et al. [6]. Most of these approaches leverage image features (i.e., keypoints) tracking across multiple frames to estimate the camera ego-motion.

For decades, feature detection and description methods were hand-engineered to guarantee invariance to geometrical or photometric changes and robustness concerning matching procedures. Many of the state-of-the-art feature extraction algorithms have shown impressive results in many applications (Chen et al. [7], Macario et al. [8]). Starting from the pioneering work by Moravec [9], we have witnessed a rapid improvement of feature extractors, from the Harris detector by Harris et al. [10] and blob detector by Matas et al. [11], to the famous SIFT by Lowe et al. [12], FAST by Rosten et al. [13] and SURF by Bay et al. [14].

More recently, the ORB extractor by Rublee et al. [15] has become quite popular as the pillar of the famous ORB–SLAM framework by Mur-Artal et al. [16], which has been further improved in its subsequent versions, i.e., ORBSLAM2 (Mur et al. [17]) and ORBSLAM3 (Campos et al. [18]).

Choosing the feature extractor that best suits the application at hand is, in general, not trivial. Each method might be appropriate for a specific condition (e.g., illumination or blur degree) or scenarios (e.g., automotive, indoor, aerial). In addition, these approaches rely on many hyper-parameters having optimal values which are hardly a priori known and could change significantly from one context to another.

Viable alternatives to hand-engineered feature extractors have recently been proposed through the exploitation of data-driven representation learning approaches (Jing et al. [19]). Deep learning is widely utilized for this objective, and new techniques based on both supervised and self-supervised approaches are continuously being developed. Models based on Convolutional Neural Networks (CNNs), in particular, are capable of computing features that exhibit robust invariance to geometric and photometric changes, including illumination, background, viewpoint, and scale (Wang et al. [20]).

A pioneering contribution to the field of learned features and descriptors was made by Verdie et al. [21], which lay the foundations for learning-based keypoints and descriptors detection by proposing a simple method to identify potentially stable points in training images, and a regression model that detects keypoints invariant to weather and illumination changes. Lenc et al. [22], instead, proposed DetNetm, which learns to compute local covariant features. This approach was improved by Zhang et al. [23] with TCDET, which ensures the detection of discriminative image features and guarantees repetitiveness. Finally, Barroso-Laguna et al. [24] fused hand-engineered and learned filters to obtain a lightweight and efficient detector with real-time performance.

The aforementioned methods treat feature detection and description as separate problems. In the last few years, strategies that jointly detect features and compute the associated descriptors are spreading. LIFT by Yi et al. [25] was among the first attempts in this direction and was followed by other successful models, including Superpoint by Detone et al. [26], LF-Net by Ono et al. [27] and D2-Net by Dusmanu et al. [28].

While the previous works propose general purpose algorithms, recent researchers have steered attention toward the integration of learning-based feature extractors into VO and SLAM systems. DXSLAM by Li et al. [29] exploits CNN-based features to obtain scene representations that are robust to environmental changes and combines them into the SLAM pipeline. To meet the high demand for neural network models for graph data, GCN (Graph Convolutional Neural Networks) was introduced by Derr et al. [30]. GCN approaches bring significant improvement in a variety of network analysis tasks, one of which is learning node representations. In particular, they were successfully used by Tang et al. in [31] and in its improved version, GCNv2 (Tang et al. [32]), for the task of keypoint and descriptor extraction.

DROID–SLAM by Teed et al. [33], relies, instead, on the end-to-end paradigm and is characterized by a neural architecture that can be easily adapted to different setups (monocular, stereo and RGB-D).

Inspired by the discussions above, this work intended to provide a benchmark study for researchers and practitioners that highlights the differences between learning-based descriptors and hand-engineered ones in visual SLAM algorithms. For this purpose, we selected one of the most popular visual SLAM algorithms, i.e., ORBSLAM3 by Campos et al. [18], and studied how the localization performance changed when its standard feature extraction pipeline was replaced with learning-based feature extractors. Namely, we employed Superpoint as the CNN-based keypoint detector and descriptor extractor. The two versions of ORBSLAM3 were compared on three different reference datasets widely employed in the robotics community. Different metrics were considered to provide an extensive quantitative analysis.

An attempt to combine Superpoint with ORBSLAM2 was also made by Deng et al. [34]. However, their analysis was limited to a single dataset, without studying the impact that the different hyper-parameters of ORBSLAM2 had on the overall performance. Instead, we propose an in-depth benchmark that includes and discusses localization performance and memory\computational resources consumption and compares the two ORBSLAM3 versions, (the first with Superpoint and the second with standard hand-engineered ORB features), under several hyper-parameter configurations.

### Contribution

As highlighted by the literature review, in recent years learned features have emerged as a promising alternative to traditional handcrafted features. Despite their demonstrated robustness to image non-idealities and generalization capabilities, to the best of our knowledge, there has been limited research directly comparing standard and handcrafted features in visual odometry applications. Thus, we believed it was crucial to conduct a thorough benchmark between learned and handcrafted features to assess their relative strengths and limitations in the context of visual odometry and SLAM.

This study could be beneficial to inform future development efforts, guiding the design and implementation of more effective and efficient algorithms. Moreover, comparing the performance of learned and handcrafted features on a diverse range of datasets provides a better understanding of the generalization capabilities of the approaches and the applicability of the algorithms to real-world scenarios.

To summarize, our contributions are as follows:A study on the integration of learned features into the ORBSLAM3 pipeline.A thorough evaluation of both learned and hand-crafted features across three diverse datasets, and considering different application domains.A performance comparison between learned and handcrafted features in terms of computational efficiency and execution timing.

The present study is structured as follows. Section 2 provides a comprehensive overview of the algorithms employed in the work and summarizes the main contribution of our work. The methodology and implementation process are explained in detail in Section 3. The results of the experiments carried out in this study are presented in Section 4, and their implications are discussed and analyzed in Section 5.

Finally, Section 6 concludes the study by offering insights that are valuable for future research and development.

## 2. Background

The high-level pipeline of a SLAM system can be divided into two main blocks: front-end and back-end. The front-end block is responsible for feature extraction, tracking, and 3D triangulation. The back-end, on the other hand, integrates the information provided by the front-end and fuses the IMU measurements (in the case of VIO approaches) to update the current and past pose estimates.

The aim of this work was to evaluate the performance of the overall SLAM pipeline when the hand-engineered feature extractors of the front-end were replaced with learning-based ones. More specifically, as mentioned in the previous section, we integrated Superpoint in the front-end of the ORBSLAM3 framework and compared this configuration against the standard one. Therefore, in the following, we briefly summarize the main characteristics of the two methods, while in Section 3 we describe the changes made to the ORBSLAM3 pipeline in order to perform the integration with Superpoint.

### 2.1. ORBSLAM3

ORBSLAM3 (Campos et al. [18]) has become one of the most popular modern keyframe-based SLAM systems, thanks to the impressive results shown in different scenarios. It can work with monocular, stereo, and RGB-D camera setups and can be used in visual, visual–inertial, and multi-map SLAM settings.

When ORBSLAM3 is used in the monocular setup-based configuration, we can identify three main threads that run in parallel:The Tracking thread receives, as input, the current camera frame and outputs an estimated pose. If the incoming frame is selected as a new keyframe, this information is forwarded to the back-end for global optimization purposes. In this stage, the algorithm extracts keypoints and descriptors from the input images using the ORB feature detector and the BRIEF feature descriptor. Moreover, the algorithm matches the keypoints and descriptors from the current image to those from previous images. This stage uses the global descriptor index, a data structure that allows the efficient matching of features across multiple images. A prior motion estimation is also performed in this thread.The Local Mapping thread handles the insertion and removal of map points and performs map optimization. The local mapping thread is responsible for incorporating new keyframes and points, pruning any unnecessary information, and improving the accuracy of the map through the visual (or visual–inertial) bundle adjustment (BA) optimization process. This is accomplished by focusing on a small group of keyframes near the current frame.The Loop & Map Matching thread identifies loop closings. If a loop is detected, a global optimization procedure is triggered to update the map points and poses to achieve global consistency.

ORBSLAM3 employs ORB (Rublee et al. [15]) as a feature extractor, which relies on FAST (Rosten et al. [13]) for keypoint detection and BRIEF by Calendor et al. [35] to compute a rotation invariant 256-bit descriptor vector. ORB is a fast detector that extracts features that exhibit different invariances, such as viewpoint and illumination, and has high repeatability. This makes it well-suited for challenging environments, where the camera motion is fast and erratic. On the other hand, ORB features are not completely scale-invariant and are sensitive to orientation changes.

To overcome these limitations, in the ORBSLAM3 implementation, the ORB feature extraction process leverages an image pyramid strategy: multiple scaled versions of the same frame are used to compute features at different image resolutions. While this improves the robustness with respect to scale variations, it entails a higher computational cost.

In this work, we replaced the ORB feature extractor with the Superpoint one in the ORBSLAM3 pipeline to assess the benefits and the limitations of sparse learning-based features for pose estimation applications. Therefore, in the following, we describe the characteristics of Superpoint before providing details on its integration with ORBSLAM3.

### 2.2. Superpoint

The Superpoint learning model is based on a self-supervised strategy able to jointly extract keypoints and relative descriptors. It exploits a convolutional encoder–decoder architecture to extract features and descriptors from two different learning pipelines. Specifically, the Superpoint pipeline (Figure 1) receives a colored 3-channels image, which is then converted into a 1-channel grayscale image with dimensions H×W×1 (where *H* and *W* are the height and the width of the image in pixels, respectively) and outputs a dense H×W map, having pixels values which express the probability of being a Superpoint. The descriptor decoder pipeline, instead, computes a unique 256 element descriptor for each detected keypoint.

Superpoint takes advantage of a homographic adaptation strategy during the learning phase to achieve robustness to scale and geometric scene structure changes without the need to have ground truth keypoints and descriptors. In addition, by exploiting GPU parallelism, the evaluation phase of the algorithm is fast and compatible with the real-time constraints of most applications.

## 3. Superpoint Integration with ORBSLAM3

This section details the integration of the Superpoint feature extraction pipeline with the front-end of the ORBSLAM3 framework. Specifically, we replace the ORB extraction module of ORBSLAM3 with the keypoints and descriptors computed by feeding input frames in the Superpoint network.

In Figure 1 the implemented pipeline is represented; specifically, the learned encoder sub-block of the Superpoint convolutional encoder–decoder architecture (sub-block A in the Figure 1) is used to extract robust visual cues from the image, and consists of convolutional layers, spatial downsampling via pooling and non-linear activation functions. The encoder input, represented by the H×W-sized image, is then converted to a $\frac{H}{8}\times \frac{W}{8}$ feature map after the network convolutional layers.

Features and descriptors are then extracted through two different non-learned decoder pipelines, namely the interest point decoder and the descriptor decoder, respectively. Both decoders receive as input M×N×1 images. The interest point decoder outputs a M×N×1 matrix of “Superpointness” probability. On the other hand, the descriptor decoder outputs the associated M×N×D (with D=256) matrix of keypoint descriptors.

Both decoders operate on a shared, and spatially reduced, representation of the input and use non-learned up-sampling to bring the representation back to H×W.

The integration of Superpoint into the ORBSLAM3 back-end is not a direct process. This is due to the fact that the FAST detector in ORBSLAM3 was specifically designed to operate across multiple pyramid levels and various image sub-blocks, to ensure an evenly distributed keypoint arrangement. Superpoint, instead, processes the full-resolution image and outputs a dense map of keypoint probabilities and associated descriptors. Therefore, we modified the ORBSLAM3 sub-blocks extraction methods to meet this specification.

To allow Superpoint to process the image at multiple pyramid levels similarly to ORB, we scaled the input frame according to the number of levels required and forwarded each scaled image through the neural network independently. Concerning the image sub-block division, we noticed that Superpoint could intrinsically extract features over the entire image. Therefore, differing from ORBSLAM3, feature sparsity could be achieved without additional image processing strategies, such as sub-cell division. Instead, since Superpoint computes a dense keypoint probability map, we thresholded it to select the *K* features with the highest probability. This procedure enabled the control of the number of keypoints extracted and imposed constraints on both the Superpoint and ORB extractors by ensuring a consistent number of features.

Differing from the ORB descriptors, which are characterized by a 256-bit vector, Superpoint provides vectors of float values that cannot be matched with the Hamming-based bit-a-bit distance of ORBSLAM3. Instead, for Superpoint features, we adopted the L2-norm between descriptors vectors and tuned the matching thresholds accordingly.

Finally, we also adapted the loop & map matching thread of ORBSLAM3 to use Superpoint features. In ORBSLAM3, the BoW representation is used to match local features between images, allowing the system to identify common locations across different frames. This approach is efficient and effective for place recognition in ORBSLAM3, as it allows for quick matching between images, and robust recognition of similar environments, despite changes in lighting, viewpoint, and other factors. By utilizing the BoW representation, ORBSLAM3 can maintain an accurate and consistent mapping of environments, even in challenging conditions. For this purpose, we used the Bag-of-Words (BoW) vocabulary made available by Deng et al. [34]. The Bag-of-words vocabulary is trained with the DBoW3 library (https://github.com/rmsalinas/DBow3, [36] accessed on 15 October 2022) on Bovisa_2008-09-01 (Bonarini et al. [37], Ceriani et al. [38]) with both outdoor and mixed sequences.

The full pipeline from input frames to output pose estimation can be summarized as follows:1.Image pre-processing: the image is resized and, if needed, rectified to remove lens distortion. The resulting frame is then converted into a single-channel grayscale format.2.Feature detection and description: the pipeline detects and describes salient features in the image using the Superpoint feature detector and descriptor. The feature extraction operation is performed by taking into account several scale levels of the input image to enhance scale invariance and increase the number of detected features.3.Keyframe selection: the system selects keyframes from the input images, which are frames deemed to be representative of the environment.4.Map initialization and update: the system uses the information from the keyframes to initialize a map of the environment and continually updates this map as new frames are processed.5.Motion estimation: the pipeline uses monocular visual odometry motion estimation algorithms to estimate the relative motion between the current and the previous frame.6.Pose estimation: the system integrates the motion estimates to compute the absolute pose of the camera in the environment.7.Loop closure detection: the pipeline regularly checks for loop closures, which occur when the camera revisits a previously visited location. If a loop closure is detected, the pipeline updates the map and refines the pose estimates.

In the following, to distinguish the standard ORBSLAM3 pipeline from the one that integrates Superpoint, we refer to the latter as SUPERSLAM3.

## 4. Experiments

In this section, we describe the experimental setup and discuss the results. Performance and resource utilization comparisons were designed to investigate the different effects of the Superpoint and standard ORB feature extraction pipelines on the ORBSLAM3 back-end.

We performed two different types of analysis: a performance analysis suited to comparing the localization accuracy achieved with ORB and Superpoint feature extractors, and a computational analysis that examined the memory and resource usage of the GPU and CPU. Finally, we provide and discuss the impact that the standard ORB extractor and the Superpoint network had on the execution times of the ORBSLAM3 feature extraction and feature matching blocks.

This work mainly focused on the comparison of deep feature extractor against ORB. Therefore, we used the monocular configuration of ORBSLAM3, although the considerations we drew could be extended to the stereo, visual–inertial, and RGB-D cases.

To provide an extensive and in-depth analysis, multiple datasets from different domains (aerial, automotive, and hand-held) and scenarios (outdoor, indoor) were selected. To compare ORBSLAM3 and SUPERSLAM3 in hand-held camera scenarios, we considered the TUM-VI by Schubert et al. [39] dataset. It offered numerous challenges, including blur due to camera shaking, six degrees of freedom (6-DoF) motions, and rapid camera movements. To evaluate the algorithms in automotive and aerial scenarios, we chose the KITTI by Geiger et al. [40] and the EuRoC by Burri et al. [41] datasets, which, in general, were characterized by sudden appearances and photometric changes, and were subject to various motion constraints. Specifically, we ran tests on all the six *Vicon Room* and five *Machine Hall* sequences of EuRoC (for a total of 27,049 frames), and on ten sequences (from *00* to *10*) of the KITTI dataset (for a total of 23,201 frames). On the other hand, for the TUM-VI we selected only six *Room* sequences (from *Room_1* to *Room_6*) (for a total of 16,235 frames), where ground truth poses estimated by the motion capture system were available for the entire trajectory.

### 4.1. Parameter Exploration

The original ORBSLAM3 implementation relies on different parameters that control tracking, loop closure, and matching behaviors. Among all of these, we focused on the two with the highest impact on the system performance, i.e., *nFeatures* and *nLevels*. Specifically, we considered several combinations of the latter parameters to assess how they affected the performance of the ORBSLAM3 and SUPERSLAM3 pipelines.

For both the algorithms analyzed, *nFeatures* defines the number *N* of features extracted from an image. In SUPERSLAM3, this aspect is controlled by extracting the *N* features with the highest Superpoint probability value. Instead, in ORBSLAM3, *N* represents the maximum number of features that are extracted for each level of the image pyramid scale. *nLevels* describes, for both the approaches, the numbers of pyramid scale levels processed in the feature extraction pipeline, andi.e., the number of times a frame is scaled before computing features and descriptors.

Other parameters introduced in the original ORBSLAM3 implementation are mostly dataset-dependant and, therefore, for both SUPERSLAM3 and ORBSLAM3 we experimentally found the best possible values and kept them unchanged during all tests. We also decided not to change the *scaleFactor* parameter.

Similar considerations were made for the *(init/min)ThFAST* FAST threshold parameters, with the exception that they were meaningless in SUPERSLAM3 and, therefore, considered only in the ORBSLAM3 evaluations.

### 4.2. Experimental Setup

The comparisons were performed by running ORBSLAM3 and SUPERSLAM3 on the sequences previously listed. For each sequence, multiple parameter configurations were analyzed by changing the number of pyramid scales used for image scaling (i.e., *nLevels*) within the range of [1, 4, 8] and the number of keypoints extracted (i.e., *nFeatures*) within the range of [500, 750, 900, 1000].

The intervals for the parameters *nLevels* and *nFeatures* were established with the consideration that both ORBSLAM3 and SUPERSLAM3 often encounter initialization problems in most sequences when the number of features is below 500. On the other hand, increasing the number of features beyond 1000 does not result in improved performance for the algorithms and instead leads to a decrease in computational efficiency.

Additionally, we conducted experiments on a subset of sequences, the images of which were deliberately blurred to evaluate the performance of the algorithms on non-ideal inputs.

In particular, we selected the *MH_02* sequence from the EuRoC dataset, the *room_6* sequence from the TUM-VI dataset, and the *07* sequence from the KITTI dataset and applied a Gaussian blur filter to their images with a patch size of 12 pixels and a standard deviation of 2.15 pixels in both directions (as depicted in Figure 2). The parameters for this set of experiments were based on the analysis performed on the standard sequences.

For *MH_02*, we set *nFeatures* to 700 and *nLevels* to 8. For *room_6* and *07*, *nFeatures* is set to 900 and *nLevels* to 4.

### 4.3. Evaluation Metrics

#### 4.3.1. Pose Estimation Metrics

To quantitatively assess the accuracy of both approaches, we utilized various commonly used metrics.

In particular, we considered the absolute pose error (APE), which is composed of rotational (expressed in degree) and positional (expressed in meters) parts. The absolute pose error $E_{i}$ is a metric for investigating the global consistency of a trajectory. Given the ground-truth poses $P_{ref}$ and the aligned estimation $P_{est}$, we can define as $X_{ref,i}$ and ${\hat{X}}_{est,i}$, respectively, the i−th pose point of the ground truth and the estimated trajectories. The APE error can then be evaluated as the absolute relative pose between the estimated pose (Equation 1) and the real one (Equation 2) at timestamp *i*.
(1)$${\hat{X}}_{est,i}\in P_{est}\;0\leq i\leq N$$
(2)$$X_{ref,i}\in P_{ref}\;0\leq i\leq M\;\mathrm{and}\;M\leq N$$
where *N* and *M* are respectively the numbers of poses in the ground truth and the estimated trajectories. The APE statistic can be easily calculated from the relation (Equation 3).
(3)$$E_{i}=P_{est,i}⊖P_{ref,i}=P_{ref,i}^{-1}P_{est,i}\in SE\left(3\right)$$

It can be decomposed into its translational (Equation 4) and rotational components (Equation 5).
(4)$$APE_{tr,i}=||trans\left(E_{i}\right)\left|\right|$$
(5)$$APE_{rot,i}=\left|\left(angle\left(\log_{SO(3)}\left(rot\left(E_{i}\right)\right)\right)\right)\right|$$
where $E_{i}$ belongs to the special Euclidean group of 3D rotations and translations, trans(.) is a function that extracts the translation component of a pose, rot(.) extracts the rotational matrix from the $E_{i}$ pose matrix, and angle(.) is a function that extracts the rotation angle from a rotation matrix. The exact form of the angle(.) function may vary, depending on the convention used for the rotational part representation.

Specifically, the logarithm of $rot(E_{i})$ gives the axis and angle of the relative rotation, which can then be converted into a scalar angle using the angle(.) function.

|.| and ||.|| are the absolute value and the Euclidean norm operators, respectively.

To measure the overall quality of the trajectory, we used the Root Mean Square Error (RMSE) of the Absolute Pose Error (APE), which was further divided into the Absolute Trajectory Error (ATE, as described in Equation (Equation 6)) and the Absolute Rotational Error (ARE, as described in Equation (Equation 7)). For simplicity, we refer to the RMSE values of ATE and ARE as ATE and ARE, respectively.
(6)$$ATE=\sqrt{\frac{1}{N}\sum_{i=0}^{N-1}{||APE_{tr,i}\left|\right|}^{2}}$$
(7)$$ARE=\sqrt{\frac{1}{N}\sum_{i=0}^{N-1}{||APE_{rot,i}\left|\right|}^{2}}$$

ATE and ARE are compact metrics to evaluate the position and rotation estimations and provided an immediate and quantitative measure to compare the tracking algorithms.

To generate the evaluation metrics, we used EVO (Grupp et al. [42]). However, EVO only considers the correspondence between the estimated trajectory and ground truth based on the timestamps, which may result in inaccurate outcomes and incorrect conclusions. This is because both SUPERSLAM3 and ORBSLAM3 may lose tracking and produce fewer poses than those provided by the ground truth.

Therefore, in our analysis, we chose to also include the Trajectory Ratio metric (TR, as defined in Equation (Equation 8)), along with ATE and ARE, to evaluate the proportion of estimated poses relative to the total number of ground truth poses:(8)TR=(M/N)∗100.

#### 4.3.2. Memory Resource and Computational Metrics

We analyzed the computational statistics of both ORBSLAM3 and SUPERSLAM3, based on two main aspects:Resource analysis, including the average allocated memory for CPU and GPU (both expressed in MB) and the utilization of computational resources (expressed as a percentage of the total available resources).Time analysis (in milliseconds ms) for the main functional blocks of SUPERSLAM3, including the feature extraction module and the descriptor matching module.

We evaluated the average CPU and GPU memory allocation and computational resource utilization for all combinations of parameters. In addition, we conducted the time analysis by considering the average extraction and matching times for a 512 × 512 image as a reference. We evaluated the matching time, based on an average of 200 matched features. On the other hand, the extraction time depended on the values selected for *nFeatures* and *nLevels*. Hence, we provided time statistics for *nFeatures* = 1000 and *nLevels* = 1 both for SUPERSLAM3 and ORBSLAM3.

### 4.4. Implementation and Training Details

For the SUPERSLAM3 tests, we used a set of pre-trained network weights, i.e., the original COCO-based training weight file provided by the Magic Leap research team (Detone et al. [43]). As the author states in the original paper (Detone et al. [26]), the SuperPoint model is first trained using the Synthetic Shapes Dataset and then refined using the 240 × 320 grayscale COCO generic image dataset (Lin et al. [44]) by exploiting the homographic adaptation process.

We ran all our tests using a Nvidia GeForce RTX 2080 Ti with 12.0 GB of dedicated RAM and an Intel(R) Core(TM) i7-9800X CPU @ 3.80 GHz 3.79 GHz with 64.0 GB of RAM.

## 5. Results and Discussion

For the purpose of clarity in presenting the results, we adopted the following compact notation to represent the dataset and its corresponding set of parameter configurations: *dataset-name_sequence-name_nFeatures-value-nLevels-value*.

Based on the performance results, presented in Table 1, Table 2 and Table 3, we observed varying trends in the performance of ORBSLAM3 and SUPERSLAM3. While some sequences showed good tracking performance, in terms of ATE, ARE, and TR for both algorithms, there were others in which SUPERSLAM3 failed to initialize (indicated by *fail* in Table 1, Table 2 and Table 3). Conversely, there were also sequences where ORBSLAM3 was outperformed by SUPERSLAM3.

More specifically, in EuRoC (see Table 1) the results of both algorithms were comparable for all of the medium and low complexity sequences (MH_01, MH_02, MH_03, V1_01). However, in complex scenes, the performance of SUPERSLAM3 dropped, while ORBSLAM3 maintained a reasonable estimation accuracy. In our opinion, to explain this different behavior, it is important to note that EuRoC included sequences recorded with flying drones into indoor environments, which resulted in images affected by non-idealities specifically related to fast motions and poor illumination. While in some instances, SUPERSLAM3 performed slightly better than ORBSLAM3 (e.g., for EuRoC_MH_05_700_8), upon visual inspection we noticed that Superpoint failed to detect keypoints in scenes with high levels of blur (e.g., due to rapid motion or rapid panning).

The KITTI dataset (Table 2) included outdoor automotive scenarios and had the highest number of sequences in which the algorithms tended to experience initialization failures. This was the case, for example, for SUPERSLAM3 when configured with *nFeatures* = 500 and *nLevels* = 1. We believe that most of the difficulties were related to the low amount of texture and distinctive cues in the sequences, e.g., due to the high portion of the images characterized by sky or asphalt, which reduced the number of informative features that could be detected. In general, from our understanding, this caused the results on KITTI to be worse than those obtained with the other two datasets, both for ORBSLAM3 and for SUPERSLAM3.

The TUM-VI dataset (see Table 3) included handheld scenes from an indoor environment and was the only dataset where both the algorithms provided reasonable performance across all sequences and parameter configurations. From a quantitative point of view, performance was generally comparable, with ORBSLAM3 achieving slightly higher metrics.

In Figure 3, we present three qualitative plots depicting the estimated trajectories of three sample sequences. By comparing these plots with the quantitative results in Table 1, Table 2 and Table 3, we observe that, on EuRoC_MH_05_700_8, SUPERSLAM3 outperformed ORBSLAM3 in terms of ATE and ARE. The trajectory ratio was comparable, indicating that both algorithms never lost the position tracking. On the other hand, in TUM_room_4_1000_8, SUPERSLAM3 performance was significantly worse than ORBSLAM3, with ATE and ARE values of 1.01 and 26.70 for SUPERSLAM3, and 0.08 and 2.48 for ORBSLAM3, respectively. The third trajectory from the KITTI dataset showed poor performance for both algorithms, particularly with regard to the ATE metric.

As expected, the results suggested that, as the number of features and pyramid levels increased, the values of TR increased, and ATE and ARE values decreased. In general, a larger number of features and pyramid levels could potentially improve the accuracy of the estimated trajectories, although the computational cost increased. It is worth noting that none of the trajectories estimated by SUPERSLAM3 had notably higher accuracy than those provided by ORBSLAM3. However, to better understand these results it is important to note that the set of Superpoint weights used in SUPERSLAM3 was trained on the COCO dataset. Hence, the detected keypoints were not specifically designed for automotive, handheld, or aerial applications. Therefore, the fact that SUPERSLAM3 could achieve performance comparable to ORBSLAM3 in most of the sequences was remarkable and showed its robustness and generalization capabilities.

On the blurred sequences, SUPERSLAM3 often lost feature tracking. This was particularly evident in the KITTI dataset, where turnings and curves were, in most cases, poorly estimated due to directional blur effects. This can also be observed in Table 4 which shows how ORBSLAM3 achieved low errors, even on blurred sequences, while SUPERSLAM3’s performance dropped significantly. We believe that this phenomenon was related to the absence of blurred images in the homographic adaptation technique used in the training process of the Superpoint network.

Computational analysis results are presented in Table 5 and Table 6 for resource and time analysis, respectively. It can be observed that the average CPU memory utilization was higher for SUPERSLAM3 compared to ORBSLAM3. This could be ascribed to the larger number of detected features stored during the tracking process. In particular, both Superpoints and ORB features were represented by 256-element vectors. However, while each element of the ORB vector was represented by a binary value, the Superpoint descriptor contained 64-bit floats, which led to higher memory usage. The average GPU utilization of SUPERSLAM3 was mainly dependent on the number of pyramid stages that needed to be forwarded through the network and remained almost constant when changing the values of both the *nFactor* and *nLevel* parameters. Indeed, Superpoint computed keypoints and descriptors for the entire image in a single forward pass, repeated for each level of the pyramid ladder. On the other hand, the average GPU memory was mainly used to store network weights.

According to a time statistics analysis, the computation of the Superpoint feature and descriptor was faster than that of the ORB keypoint and descriptor. Specifically, Table 6 shows that the average extraction time for ORBSLAM3 was approximately double that of the Superpoint descriptor. In contrast, the feature matching time for Superpoint features was higher than that of ORBSLAM3. This was expected since the feature matching process in SUPERSLAM3, which utilized the L2 norm, was slower than ORBSLAM3’s bit-wise descriptor matching method.

## 6. Conclusions

State estimation and tracking are fundamental in robotics and automotive applications, as they enable high-precision real-time SLAM systems for navigation tasks. These tasks require the selection of high-quality keyframes across images for accurate tracking, which can be achieved through the use of single-camera applications.

In addition to traditional methods, there has been a surge in the use of learning-based methods, which can automatically learn robust feature representations from large datasets and simultaneously estimate feature keypoints and descriptors with low computational costs and strong generalization capabilities.

In this study, we integrated Superpoint features within the ORBSLAM3 pipeline. We then presented a quantitative evaluation of the tracking and computational performance of the integration of Superpoint learned features into the ORBSLAM3 pipeline (i.e., SUPERSLAM3). We tested both SUPERSLAM3 and ORBSLAM3 on three datasets from different domains, including aerial, automotive, and handheld.

Our findings indicated that learned features could achieve good pose estimation results. However, by analyzing the results obtained, we hypothesized that including blurry image patterns and rotations in the training phase could enhance the system’s robustness and reliability. Training on a larger dataset could also enhance the extraction of robust Superpoint features, while increasing the generalization capabilities of the overall algorithm.

In our computational analysis, we observed that SUPERSLAM3 had faster performance for keypoints and descriptors extraction compared to ORBSLAM3. However, it was slower in the features matching phase.

Future work could consider the use of learned features trained on a large dataset to improve generalization capabilities and overall performance in terms of tracking estimation. The Superpoint matching phase could be enhanced through the integration of a GPU-based matching process, such as SuperGLUE (Sarlin et al. [45]). Finally, based on our experimental results, we concluded that incorporating artificially blurred and non-ideal images into the training process of the network could enhance the robustness and consistency of the detector.

## Figures and Tables

**Figure 1 sensors-23-02286-f001:**
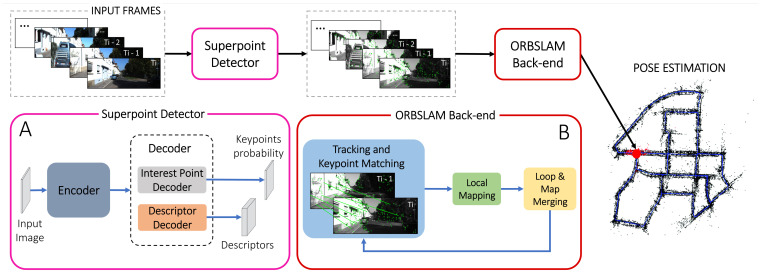
The figure depicts the integration of the Superpoint feature detector and descriptor into the ORB-SLAM3 pipeline. The traditional ORB and BRIEF feature detection and description methods have been replaced with the Superpoint pipeline. The input images are transformed into the grayscale format and fed into the Superpoint detector pipeline (**A**). The Superpoint encoder–decoder pipeline is composed of a learned encoder, based on several convolutional layers, and two non-learned decoders for the joint features and descriptors extraction. The detected features are then processed by the ORBSLAM3 back-end, which consists of three main components that operate as three parallel threads: the Tracking, Local Mapping, and Loop & Map Merging threads (**B**). The back-end extracts keyframes, initializes and updates the map, and performs motion and pose estimation, both locally within the Local Mapping Thread and globally within the Loop & Map Merging thread. If a loop closure is detected, the pose estimation is further refined.

**Figure 2 sensors-23-02286-f002:**
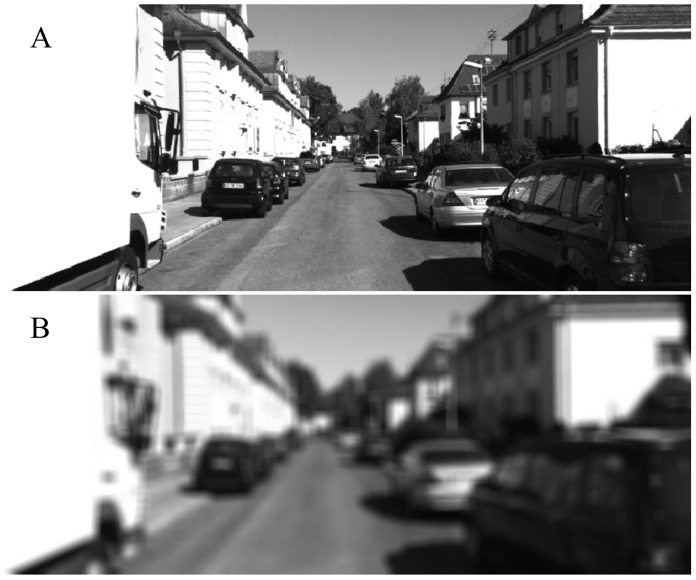
Sample image from the original (**A**) and blurred (**B**) KITTI dataset (sequence 07). The original image was blurred using a gaussian filter with a patch size of 12 pixels and a standard deviation of 2.15 pixels 241 in both directions.

**Figure 3 sensors-23-02286-f003:**
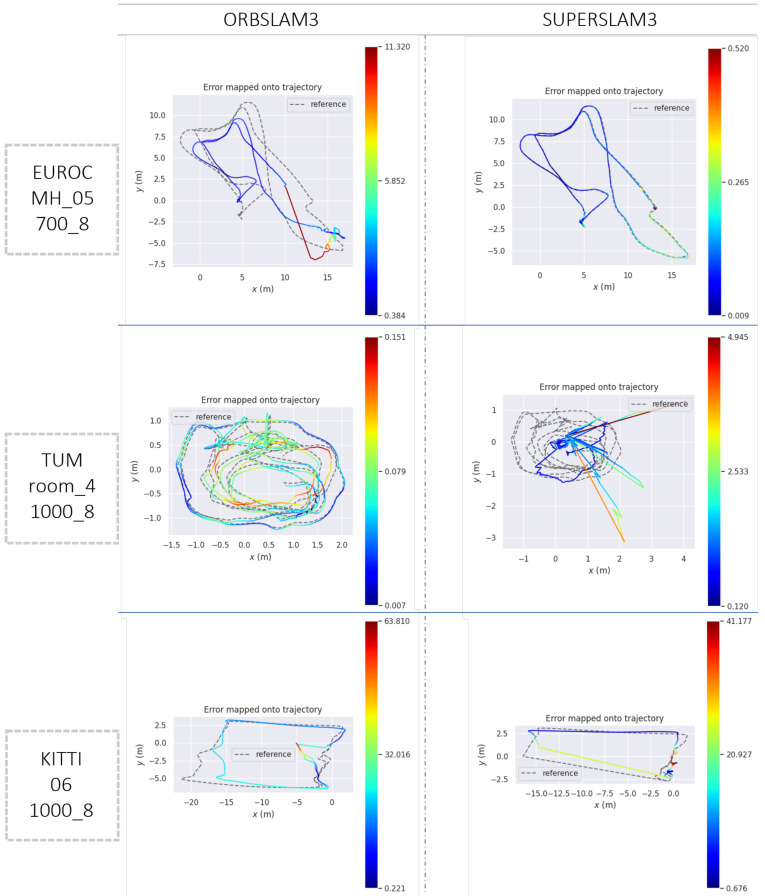
Sample trajectories comparisons using EVO tools shows that, while SUPERSLAM3 outperformed ORBSLAM3 in terms of ATE and ARE for the EuRoC MH_05 sequence with *nFeatures* and *nLevels* set to 700 and 8, respectively, it performed significantly worse than ORBSLAM3 for the room_4 sequence (TUM-VI) with *nFeatures* and *nLevels* set to 1000 and 8, respectively. Both algorithms also showed poor performance for the KITTI dataset, particularly in terms of ATE.

**Table 1 sensors-23-02286-t001:** Quantitative results on EuRoC. For each sequence and for each performance metric, we highlight in bold the parameter configuration that attained the best result (for both ORBSLAM3 and SUPERSLAM3) individually for each analyzed metric (as described in the table’s right header). To indicate the one with the highest performance between the two, we color the corresponding cell and highlight the algorithm name in gray. In the table’s top header the parameters set are represented in the form ‘*nFeatures_nLevels*’.

Sequence	Algorithm	500_1	500_4	500_8	700_1	700_4	700_8	900_1	900_4	900_8	1000_1	1000_4	1000_8	
MH_01	SUPERSLAM3	*fail*	0.06	**0.04**	0.30	1.52	0.05	0.56	0.08	0.09	0.24	0.10	0.18	ATE (m)
	ORBSLAM3	**0.03**	**0.03**	0.06	**0.03**	**0.03**	0.04	0.09	0.04	0.04	0.04	0.04	0.04	
	SUPERSLAM3	*fail*	2.57	**1.89**	2.11	12.32	2.15	2.65	2.47	2.80	2.12	2.32	2.00	ARE (deg)
	ORBSLAM3	2.06	**1.9**	1.93	2.07	2.02	2.19	2.39	1.99	2.20	2.05	1.94	2.09	
	SUPERSLAM3	0.00	55.38	93.89	84.55	89.63	88.65	86.09	99.27	**99.97**	85.09	88.10	87.53	TR (%)
	ORBSLAM3	**99.97**	99.95	99.95	**99.97**	99.95	99.95	**99.97**	99.95	99.95	**99.97**	99.92	**99.97**	
MH_02	SUPERSLAM3	0.04	0.07	0.06	0.13	0.04	**0.03**	0.33	0.06	**0.03**	0.32	2.73	0.06	ATE (m)
	ORBSLAM3	0.04	0.04	0.04	**0.03**	0.04	**0.03**	**0.03**	**0.03**	**0.03**	**0.03**	**0.03**	**0.03**	
	SUPERSLAM3	1.81	1.87	1.98	1.89	1.76	**1.71**	2.03	1.79	1.76	4.05	17.03	1.82	ARE (deg)
	ORBSLAM3	1.43	1.64	1.54	**1.37**	1.68	1.51	1.62	1.66	1.80	1.66	1.77	1.73	
	SUPERSLAM3	38.45	38.26	39.41	73.68	99.24	**99.70**	81.02	90.72	91.64	83.85	89.87	91.74	TR (%)
	ORBSLAM3	99.84	**99.93**	99.77	99.90	**99.93**	99.90	99.90	99.90	99.87	99.90	99.90	99.90	
MH_03	SUPERSLAM3	*fail*	**0.03**	**0.03**	0.49	0.06	0.05	0.22	0.11	0.06	0.08	0.14	0.05	ATE (m)
	ORBSLAM3	0.04	**0.03**	0.04	0.04	0.04	0.04	0.04	0.04	0.04	0.04	0.04	0.04	
	SUPERSLAM3	*fail*	**1.4**	1.42	1.89	1.63	1.72	2.05	1.93	1.68	1.72	1.56	1.70	ARE (deg)
	ORBSLAM3	1.74	1.62	1.68	1.59	**1.58**	1.64	1.66	1.63	1.64	1.64	1.69	1.65	
	SUPERSLAM3	0.00	25.67	26.96	83.26	98.00	97.96	84.33	**98.04**	97.81	96.74	97.85	97.85	TR (%)
	ORBSLAM3	98.00	98.00	98.00	98.00	97.96	97.89	98.00	**98.04**	98.00	98.00	98.00	97.96	
MH_04	SUPERSLAM3	**0.03**	0.55	0.05	0.10	0.08	0.06	0.12	0.08	0.44	4.00	0.22	0.22	ATE (m)
	ORBSLAM3	0.16	0.13	0.10	0.12	0.15	0.06	0.11	**0.05**	0.09	0.16	0.06	0.53	
	SUPERSLAM3	1.41	1.48	**1.33**	1.45	1.53	1.45	1.63	1.51	1.63	8.46	1.45	1.67	ARE (deg)
	ORBSLAM3	1.55	1.53	1.47	1.52	1.66	**1.38**	1.69	1.42	**1.38**	1.68	1.45	2.12	
	SUPERSLAM3	8.02	85.15	36.60	55.83	92.57	76.54	65.91	77.72	**97.44**	71.13	95.13	95.47	TR (%)
	ORBSLAM3	97.64	97.98	**98.03**	97.64	97.93	97.54	97.64	97.59	97.93	97.64	97.59	97.93	
MH_05	SUPERSLAM3	**0.08**	0.12	0.11	0.16	0.14	0.15	0.16	0.21	0.16	0.27	0.92	0.18	ATE (m)
	ORBSLAM3	0.07	0.06	0.10	0.06	**0.05**	2.37	0.06	0.08	**0.05**	0.06	**0.05**	**0.05**	
	SUPERSLAM3	1.80	**1.5**	1.62	1.81	1.78	2.32	2.15	1.82	1.93	2.43	9.26	2.05	ARE (deg)
	ORBSLAM3	1.84	1.79	1.86	1.83	1.79	9.75	1.85	**1.71**	1.75	1.77	1.76	1.77	
	SUPERSLAM3	39.73	43.11	42.19	63.22	95.16	95.07	62.82	95.03	94.68	66.26	94.50	**96.13**	TR (%)
	ORBSLAM3	96.17	96.22	95.78	96.17	78.18	**96.39**	96.17	96.35	96.22	96.17	96.35	96.35	
V1_01	SUPERSLAM3	0.12	0.10	**0.09**	**0.09**	**0.09**	**0.09**	0.20	**0.09**	**0.09**	0.20	0.29	0.28	ATE (m)
	ORBSLAM3	0.10	**0.09**	**0.09**	**0.09**	**0.09**	**0.09**	0.10	**0.09**	**0.09**	**0.09**	**0.09**	**0.09**	
	SUPERSLAM3	6.07	5.61	5.62	6.60	5.70	5.62	6.46	**5.59**	5.63	7.79	9.23	7.42	ARE (deg)
	ORBSLAM3	5.95	5.56	5.54	5.80	5.54	5.55	5.52	5.55	5.60	5.52	5.57	**5.50**	
	SUPERSLAM3	28.50	93.58	94.71	59.65	92.55	**95.60**	54.64	95.47	94.64	57.62	90.63	95.23	TR (%)
	ORBSLAM3	**96.26**	95.95	96.05	96.12	**96.26**	**96.26**	96.19	96.19	96.19	**96.26**	**96.26**	96.19	
V1_02	SUPERSLAM3	0.03	0.08	0.07	0.06	0.11	0.17	0.03	0.13	0.13	**0.02**	0.11	0.26	ATE (m)
	ORBSLAM3	0.07	**0.06**	**0.06**	0.10	**0.06**	**0.06**	0.10	**0.06**	0.07	0.09	**0.06**	0.09	
	SUPERSLAM3	1.66	1.47	**1.43**	2.99	1.77	1.64	3.65	1.49	2.54	3.08	2.24	3.71	ARE (deg)
	ORBSLAM3	1.43	1.28	1.35	1.52	**1.26**	1.30	2.16	1.27	1.29	1.41	1.28	1.90	
	SUPERSLAM3	12.22	67.19	80.29	19.88	87.54	90.53	14.33	88.65	**92.11**	8.54	80.58	85.67	TR (%)
	ORBSLAM3	88.71	94.74	83.22	93.80	94.74	89.18	93.86	94.15	94.74	94.62	**94.85**	94.80	
V1_03	SUPERSLAM3	0.05	0.33	0.04	**0.02**	0.11	0.09	**0.02**	0.10	0.50	0.03	0.04	0.17	ATE (m)
	ORBSLAM3	0.11	0.12	**0.07**	0.14	0.11	**0.07**	0.16	0.60	0.12	0.13	0.08	0.08	
	SUPERSLAM3	2.16	10.85	**1.99**	3.53	3.78	3.25	2.93	3.38	24.47	2.74	2.02	10.67	ARE (deg)
	ORBSLAM3	5.59	2.67	**1.78**	4.13	2.84	1.88	4.24	16.51	2.69	3.16	2.04	2.07	
	SUPERSLAM3	7.58	22.62	23.22	6.28	**37.13**	24.48	7.21	25.97	36.11	7.07	19.87	18.47	TR (%)
	ORBSLAM3	31.22	78.08	93.02	58.31	93.76	93.25	74.50	93.81	88.51	64.22	**94.14**	93.86	
V2_01	SUPERSLAM3	0.04	*fail*	0.64	**0.03**	0.10	0.11	*fail*	0.09	0.19	**0.03**	0.40	0.11	ATE (m)
	ORBSLAM3	0.07	**0.06**	**0.06**	0.85	**0.06**	**0.06**	**0.06**	**0.06**	**0.06**	0.24	0.11	**0.06**	
	SUPERSLAM3	2.50	*fail*	2.37	0.94	1.18	1.24	*fail*	1.45	1.28	**0.81**	3.18	1.46	ARE (deg)
	ORBSLAM3	1.27	1.08	1.15	5.57	1.19	1.22	**1.06**	1.10	1.12	8.77	1.73	1.26	
	SUPERSLAM3	17.81	0.00	73.55	28.64	92.50	**92.89**	0.00	92.72	92.68	22.72	92.59	92.81	TR (%)
	ORBSLAM3	93.29	93.64	93.64	93.33	93.64	93.64	93.42	**93.68**	**93.68**	93.64	**93.68**	**93.68**	
V2_02	SUPERSLAM3	**0.02**	0.09	0.13	0.04	0.13	*fail*	0.06	0.33	0.19	0.03	0.17	0.20	ATE (m)
	ORBSLAM3	0.06	0.09	0.07	0.06	0.06	0.11	0.63	0.07	0.06	**0.05**	0.08	0.06	
	SUPERSLAM3	**0.71**	1.26	1.47	0.95	1.63	*fail*	1.39	4.36	1.42	1.25	2.53	4.16	ARE (deg)
	ORBSLAM3	0.83	2.09	0.81	0.80	**0.76**	1.68	1.73	1.07	0.90	0.78	1.32	0.84	
	SUPERSLAM3	12.31	73.04	66.10	15.67	90.33	0.00	33.22	70.74	**93.23**	27.77	81.13	84.16	TR (%)
	ORBSLAM3	93.61	96.59	96.85	93.61	96.93	96.72	96.89	96.93	97.02	93.65	**97.23**	97.02	
V2_03	SUPERSLAM3	**0.01**	0.51	0.04	**0.01**	**0.01**	0.06	**0.01**	**0.01**	0.08	*fail*	**0.01**	0.08	ATE (m)
	ORBSLAM3	**0.06**	0.08	0.08	**0.06**	0.09	0.32	0.08	0.09	0.34	**0.06**	0.27	0.11	
	SUPERSLAM3	2.05	25.63	2.47	3.21	2.22	3.48	4.76	3.56	2.79	*fail*	**0.66**	2.82	ARE (deg)
	ORBSLAM3	3.20	2.01	**1.79**	3.32	2.28	5.15	2.92	2.10	7.77	2.98	5.29	2.40	
	SUPERSLAM3	4.84	26.12	22.79	5.31	5.10	**28.56**	5.31	5.15	22.63	0.00	5.62	23.52	TR (%)
	ORBSLAM3	46.31	83.51	88.40	47.45	87.04	91.73	52.13	88.92	90.43	51.87	81.74	**92.92**	

**Table 2 sensors-23-02286-t002:** Quantitative results on KITTI. For each sequence and for each performance metric, we highlight in bold the parameter configuration that attained the best result (for both ORBSLAM3 and SUPERSLAM3) individually for each analyzed metric (as described in the table’s right header). To indicate the one with the highest performance between the two, we color the corresponding cell and highlight the algorithm name in gray. In the table’s top header the parameters set are represented in the form ‘*nFeatures_nLevels*’.

Sequence	Algorithm	500_1	500_4	500_8	700_1	700_4	700_8	900_1	900_4	900_8	1000_1	1000_4	1000_8	
00	SUPERSLAM3	*fail*	**0.04**	**0.04**	1.50	14.22	0.59	4.86	13.80	16.77	4.11	14.28	10.39	ATE (m)
	ORBSLAM3	0.98	0.80	1.07	1.06	8.45	11.14	**0.41**	19.51	22.41	1.87	20.07	18.66	
	SUPERSLAM3	*fail*	156.21	158.09	1.18	1.79	**0.46**	0.88	1.91	1.70	0.89	2.06	1.65	ARE (deg)
	ORBSLAM3	1.71	1.84	1.54	**0.70**	1.82	5.95	0.88	1.22	0.98	1.18	1.17	1.11	
	SUPERSLAM3	0.02	0.88	0.88	10.55	35.21	5.81	23.36	43.51	**78.22**	22.35	60.80	57.59	TR (%)
	ORBSLAM3	6.56	7.99	8.81	13.32	27.13	31.25	7.22	59.61	68.16	14.89	60.67	**82.01**	
01	SUPERSLAM3	*fail*	*fail*	*fail*	*fail*	*fail*	**0.35**	*fail*	3.34	0.57	*fail*	1.08	0.52	ATE (m)
	ORBSLAM3	1.72	2.32	*fail*	**0.29**	0.80	7.41	25.22	51.72	462.59	96.40	9.81	317.25	
	SUPERSLAM3	*fail*	*fail*	*fail*	*fail*	*fail*	**2.01**	*fail*	12.21	158.17	*fail*	129.34	2.25	ARE (deg)
	ORBSLAM3	153.02	163.07	*fail*	0.95	2.05	**0.55**	22.02	176.38	21.48	19.23	1.10	4.04	
	SUPERSLAM3	0.00	0.00	0.00	0.00	0.00	**6.36**	0.00	4.72	5.36	0.00	4.81	**6.36**	TR (%)
	ORBSLAM3	6.90	8.45	*fail*	9.36	12.72	47.96	31.43	37.15	96.55	33.15	40.96	**96.82**	
02	SUPERSLAM3	*fail*	*fail*	*fail*	**0.24**	3.15	3.39	0.25	5.03	3.04	0.49	0.38	3.00	ATE (m)
	ORBSLAM3	**0.03**	0.27	0.49	0.16	0.72	0.63	**0.03**	3.81	16.36	0.33	8.11	29.13	
	SUPERSLAM3	*fail*	*fail*	*fail*	3.14	2.89	1.39	1.29	2.25	1.87	1.51	**0.71**	1.43	ARE (deg)
	ORBSLAM3	1.76	0.85	1.02	0.82	0.77	**0.51**	0.58	1.85	0.98	1.40	0.99	3.76	
	SUPERSLAM3	0.00	0.00	0.00	1.67	8.26	**8.90**	2.62	7.53	5.17	2.30	6.84	5.45	TR (%)
	ORBSLAM3	1.33	2.53	4.48	1.82	4.03	5.26	1.93	8.41	37.44	2.06	19.95	**45.98**	
03	SUPERSLAM3	*fail*	**0.01**	**0.01**	0.08	0.02	0.17	0.03	0.91	1.00	0.02	1.00	1.02	ATE (m)
	ORBSLAM3	0.09	0.09	0.10	**0.08**	0.31	1.03	0.34	2.19	1.31	0.19	1.56	1.33	
	SUPERSLAM3	*fail*	6.45	94.10	6.49	2.00	1.73	4.50	0.59	**0.36**	4.44	0.63	0.74	ARE (deg)
	ORBSLAM3	7.30	4.95	4.75	3.30	0.50	0.44	0.46	0.32	0.36	1.81	**0.29**	0.39	
	SUPERSLAM3	0.00	5.12	5.12	16.85	13.98	18.73	11.99	27.09	**40.95**	9.11	30.46	27.47	TR (%)
	ORBSLAM3	15.86	21.35	21.72	20.47	32.21	59.18	17.85	**76.03**	60.42	27.47	59.93	60.42	
04	SUPERSLAM3	*fail*	*fail*	0.09	*fail*	**0.01**	0.03	*fail*	0.10	0.05	*fail*	0.13	0.07	ATE (m)
	ORBSLAM3	**0.01**	0.12	0.08	**0.01**	0.10	0.46	0.22	0.10	1.08	0.58	0.36	0.27	
	SUPERSLAM3	*fail*	*fail*	11.64	*fail*	16.72	140.35	*fail*	**0.45**	36.27	*fail*	5.27	18.13	ARE (deg)
	ORBSLAM3	100.47	46.20	13.87	80.30	129.05	5.57	**3.48**	10.03	31.96	135.82	23.75	13.99	
	SUPERSLAM3	0.00	0.00	14.76	0.00	16.24	21.77	8.12	31.73	30.63	8.12	**41.7**	31.73	TR (%)
	ORBSLAM3	17.34	47.23	45.39	17.71	43.91	**99.63**	38.01	40.96	**99.63**	52.03	86.72	**99.63**	
05	SUPERSLAM3	*fail*	*fail*	*fail*	*fail*	11.46	22.77	**0.51**	9.25	24.73	0.57	6.17	10.48	ATE (m)
	ORBSLAM3	**0.25**	0.80	0.26	1.00	8.86	5.84	0.37	10.59	8.96	2.61	11.00	8.84	
	SUPERSLAM3	*fail*	*fail*	*fail*	*fail*	0.99	4.18	**0.33**	1.88	3.83	0.68	1.85	1.35	ARE (deg)
	ORBSLAM3	1.15	1.07	5.88	0.88	1.86	**0.63**	1.87	0.98	1.61	0.87	1.06	1.44	
	SUPERSLAM3	0.00	1.41	0.00	0.04	50.53	50.53	17.67	61.28	**75.19**	15.86	32.20	64.90	TR (%)
	ORBSLAM3	6.41	7.28	12.42	10.50	40.20	53.39	18.73	51.21	95.36	25.24	77.58	**99.82**	
06	SUPERSLAM3	*fail*	0.18	**0.03**	*fail*	0.71	9.97	1.05	9.61	7.44	0.07	10.65	20.19	ATE (m)
	ORBSLAM3	**0.02**	1.71	0.04	0.83	2.23	14.71	1.25	0.47	25.15	0.15	28.90	25.65	
	SUPERSLAM3	*fail*	58.88	156.07	*fail*	**0.88**	1.74	1.61	2.15	1.71	1.47	3.28	2.39	ARE (deg)
	ORBSLAM3	0.71	4.88	**0.35**	2.67	2.24	1.60	2.27	0.71	2.63	2.95	4.38	2.03	
	SUPERSLAM3	0.00	3.63	3.63	0.00	11.44	45.59	10.54	34.15	56.40	14.90	**72.39**	53.86	TR (%)
	ORBSLAM3	5.27	10.81	7.90	8.63	15.08	60.94	10.81	38.60	84.47	12.72	**99.55**	99.46	
07	SUPERSLAM3	*fail*	*fail*	*fail*	**0.03**	7.72	9.60	2.13	2.98	19.55	3.17	12.98	23.93	ATE (m)
	ORBSLAM3	**0.08**	0.90	0.98	0.89	9.98	5.88	4.46	23.09	20.50	5.19	14.58	2.15	
	SUPERSLAM3	*fail*	*fail*	*fail*	**0.78**	3.57	4.45	1.59	1.14	6.40	1.96	5.06	8.37	ARE (deg)
	ORBSLAM3	2.07	1.97	1.44	2.26	4.44	2.04	2.98	8.87	7.45	2.12	5.79	**0.88**	
	SUPERSLAM3	0.00	0.00	0.00	6.27	73.39	76.39	31.97	**99.64**	98.64	43.42	98.91	98.82	TR (%)
	ORBSLAM3	15.17	18.53	20.62	18.71	73.12	47.32	41.14	99.46	99.09	44.23	79.20	**99.55**	
08	SUPERSLAM3	*fail*	**0.02**	*fail*	0.14	5.37	11.13	0.23	3.56	6.81	0.16	4.12	2.42	ATE (m)
	ORBSLAM3	0.16	0.16	0.18	**0.11**	0.61	3.25	0.18	3.76	3.96	0.18	6.25	11.45	
	SUPERSLAM3	*fail*	137.49	*fail*	**0.38**	2.62	1.81	7.44	2.53	2.89	5.56	1.08	1.38	ARE (deg)
	ORBSLAM3	3.22	1.57	38.84	4.05	2.72	1.65	9.03	1.65	1.90	8.91	**1.22**	2.07	
	SUPERSLAM3	0.02	0.98	0.02	2.46	11.94	**21.59**	4.69	9.83	12.11	4.20	13.90	12.31	TR (%)
	ORBSLAM3	2.28	3.27	3.24	2.90	8.16	13.76	4.62	14.98	16.70	4.62	20.81	**29.92**	
09	SUPERSLAM3	*fail*	*fail*	*fail*	**0.04**	1.28	0.66	0.40	11.14	6.26	0.77	6.03	3.96	ATE (m)
	ORBSLAM3	**0.01**	0.18	0.21	0.09	0.59	1.03	0.49	1.63	7.20	0.62	2.61	6.74	
	SUPERSLAM3	*fail*	*fail*	*fail*	1.44	5.82	0.47	0.87	2.84	1.49	**0.82**	1.80	2.54	ARE (deg)
	ORBSLAM3	0.46	1.25	1.03	1.23	0.66	0.75	1.44	0.49	1.52	1.17	**0.42**	1.66	
	SUPERSLAM3	0.00	0.00	0.00	3.39	8.61	16.72	8.61	38.59	**41.86**	10.37	29.67	32.24	TR (%)
	ORBSLAM3	2.83	7.54	6.79	5.78	9.43	19.04	8.17	21.12	56.69	9.37	39.03	**57.07**	
10	SUPERSLAM3	*fail*	*fail*	*fail*	0.05	0.42	**0.04**	0.06	5.74	0.23	0.06	0.09	0.28	ATE (m)
	ORBSLAM3	**0.04**	0.24	0.05	0.22	0.95	0.42	0.26	1.23	1.56	0.25	1.11	0.54	
	SUPERSLAM3	*fail*	*fail*	*fail*	1.26	**0.73**	4.27	3.00	2.64	3.43	2.59	3.01	2.06	ARE (deg)
	ORBSLAM3	1.27	1.30	6.92	1.50	0.86	0.68	1.34	0.73	1.13	1.47	0.69	**0.66**	
	SUPERSLAM3	0.00	0.00	0.00	5.08	17.74	7.58	10.07	**56.54**	14.24	9.33	14.65	21.15	TR (%)
	ORBSLAM3	7.16	8.83	9.08	9.66	21.32	18.32	10.24	24.73	**33.56**	9.91	22.23	30.47	

**Table 3 sensors-23-02286-t003:** Quantitative results on TUM-VI. For each sequence and for each performance metric, we highlight in bold the parameter configuration that attained the best result (for both ORBSLAM3 and SUPERSLAM3) individually for each analyzed metric (as described in the table’s right header). To indicate the one with the highest performance between the two, we color the corresponding cell and highlight the algorithm name in gray. In the table’s top header the parameters set are represented in the form ‘*nFeatures_nLevels*’.

Sequence	Algorithm	500_1	500_4	500_8	700_1	700_4	700_8	900_1	900_4	900_8	1000_1	1000_4	1000_8	
room_1	SUPERSLAM3	**0.07**	0.19	0.68	0.09	0.80	0.98	0.33	0.80	0.81	0.09	0.93	0.92	ATE (m)
	ORBSLAM3	**0.07**	**0.07**	**0.07**	**0.07**	0.18	**0.07**	**0.07**	0.21	0.21	**0.07**	0.14	**0.07**	
	SUPERSLAM3	**1.97**	2.25	5.51	3.27	2.73	15.60	2.05	2.47	2.87	3.14	6.75	4.54	ARE (deg)
	ORBSLAM3	2.37	2.31	**2.3**	2.38	5.27	2.33	2.38	5.98	6.20	2.38	4.52	**2.3**	
	SUPERSLAM3	16.45	95.00	95.85	45.59	**96.24**	96.17	90.75	96.21	96.21	82.77	95.82	**96.24**	TR (%)
	ORBSLAM3	**96.6**	96.35	96.46	**96.6**	**96.6**	96.53	**96.6**	96.42	**96.6**	**96.6**	**96.6**	**96.6**	
room_2	SUPERSLAM3	**0.02**	0.08	0.13	**0.02**	0.54	0.13	0.42	1.17	0.88	0.03	1.13	1.09	ATE (m)
	ORBSLAM3	0.08	**0.05**	0.06	0.06	**0.05**	**0.05**	**0.05**	**0.05**	**0.05**	**0.05**	**0.05**	**0.05**	
	SUPERSLAM3	5.34	5.22	5.09	5.18	**4.62**	5.23	8.49	43.13	9.14	5.07	18.81	8.15	ARE (deg)
	ORBSLAM3	5.33	5.10	5.02	**4.92**	5.07	5.06	5.09	5.06	5.04	5.10	5.07	5.07	
	SUPERSLAM3	38.20	96.56	96.39	73.00	98.54	98.47	93.10	98.54	98.58	73.77	98.61	**98.85**	TR (%)
	ORBSLAM3	98.58	98.40	98.51	98.58	98.75	98.54	98.58	**98.79**	98.72	98.58	98.68	98.75	
room_3	SUPERSLAM3	**0.04**	0.77	0.17	0.24	0.36	0.35	0.34	0.54	0.48	0.82	1.15	1.00	ATE (m)
	ORBSLAM3	**0.04**	**0.04**	**0.04**	**0.04**	1.18	**0.04**	**0.04**	0.14	0.05	**0.04**	0.11	0.16	
	SUPERSLAM3	3.79	9.19	4.22	4.27	3.69	4.67	**3.36**	3.76	9.39	3.56	4.31	4.14	ARE (deg)
	ORBSLAM3	3.89	3.87	**3.75**	3.92	14.25	3.76	3.88	4.79	3.76	3.94	5.09	5.58	
	SUPERSLAM3	7.62	83.98	91.46	76.99	97.59	86.67	81.21	86.88	98.79	74.26	86.60	**98.83**	TR (%)
	ORBSLAM3	98.97	98.94	98.97	98.97	99.04	98.90	98.97	99.33	98.97	98.97	99.29	**99.54**	
room_4	SUPERSLAM3	**0.03**	0.33	0.04	**0.03**	0.75	0.57	0.11	1.09	0.87	0.04	1.00	1.02	ATE (m)
	ORBSLAM3	**0.08**	**0.08**	0.10	**0.08**	**0.08**	**0.08**	0.11	**0.08**	**0.08**	**0.08**	**0.08**	**0.08**	
	SUPERSLAM3	2.18	2.57	2.19	**2.01**	9.76	6.37	3.66	17.60	5.95	2.03	22.94	26.70	ARE (deg)
	ORBSLAM3	2.54	2.49	**2.39**	2.55	2.49	2.46	3.83	2.48	2.48	2.56	2.50	2.48	
	SUPERSLAM3	11.98	60.46	22.62	12.48	**95.78**	83.44	30.43	86.18	89.63	16.88	91.43	91.07	TR (%)
	ORBSLAM3	97.22	97.98	97.04	97.22	97.49	97.44	97.08	**98.25**	97.89	97.22	**98.25**	97.98	
room_5	SUPERSLAM3	**0.04**	0.06	0.18	0.08	0.13	0.89	0.50	0.89	0.93	0.83	0.91	0.94	ATE (m)
	ORBSLAM3	0.08	0.08	0.09	0.08	0.15	0.45	**0.07**	0.08	0.08	0.08	0.08	0.09	
	SUPERSLAM3	3.09	5.29	3.92	**2.64**	4.59	21.34	9.57	5.63	27.07	21.52	17.03	9.70	ARE (deg)
	ORBSLAM3	4.27	4.29	4.21	4.21	4.77	4.68	4.23	4.19	**4.12**	4.26	4.22	4.26	
	SUPERSLAM3	7.90	26.38	68.00	53.07	69.27	97.37	87.85	**98.31**	88.13	72.57	84.12	94.03	TR (%)
	ORBSLAM3	98.45	91.64	97.68	98.45	**98.49**	97.75	98.45	98.00	98.00	98.45	98.00	98.14	
room_6	SUPERSLAM3	**0.05**	0.09	0.60	0.18	0.87	0.96	**0.05**	0.95	0.98	0.48	0.98	0.98	ATE (m)
	ORBSLAM3	**0.07**	0.08	0.08	0.08	0.08	0.08	0.08	0.08	**0.07**	0.08	0.08	**0.07**	
	SUPERSLAM3	3.11	2.95	3.06	2.48	3.80	**2.46**	2.50	15.49	3.19	4.80	7.72	5.22	ARE (deg)
	ORBSLAM3	2.95	2.91	2.91	2.92	2.92	2.92	**2.88**	2.92	**2.88**	**2.88**	2.90	2.90	
	SUPERSLAM3	25.91	85.32	96.24	77.16	98.56	98.63	41.92	**99.09**	98.56	67.68	98.56	98.63	TR (%)
	ORBSLAM3	98.60	98.60	**97.80**	98.60	98.37	98.37	98.60	98.60	98.98	98.60	98.67	98.56	

**Table 4 sensors-23-02286-t004:** Results of ORBSLAM3 and SUPERSLAM3 on a subset of blurred sequences.

Algorithm	Sequence	Params	ATE (m)	ARE (deg)	TR (%)
SUPERSLAM3	EUROC MH_02	700_8	0.27	2.50	0.40
ORBSLAM3	EUROC MH_02	700_8	0.08	1.71	0.99
SUPERSLAM3	TUM room_6	900_4	0.09	4.32	0.37
ORBSLAM3	TUM room_6	900_4	0.07	2.91	0.98
SUPERSLAM3	KITTI 07	900_4	7.00	4.07	0.5
ORBSLAM3	KITTI 07	900_4	0.07	1.81	0.15

**Table 5 sensors-23-02286-t005:** Analysis of the CPU and GPU performance statistics in relation to the parameter set used. AM: Average Memory, AU: Average Utilization. In the table’s top header the parameters set are represented in the form ‘*nFeatures_nLevels*’.

	500_1	500_4	500_8	700_1	700_4	700_8	900_1	900_4	900_8	1000_1	1000_4	1000_8
ORBSLAM3 CpuAM (MB)	748.8	780	780	811.2	904.8	873.6	904.8	936	999.4	904.8	967.2	967.2
SUPERSLAM3 CpuAM (MB)	5522.4	7644	9391.2	6021.6	7706.4	10,046.4	6177.6	8361.6	10,639.2	6614.4	8049.6	11,575.2
SUPERSLAM3 GpuAM (MB)	2695	2628	2647	2634	2682	2633	2669	2703	2578	2625	2617	2712
SUPERSLAM3 GpuAU (%)	17%	19%	20%	16%	18%	21%	17%	18%	21%	16%	18%	19%

**Table 6 sensors-23-02286-t006:** Time statistics: comparison of maximum and average time performance for feature extraction and descriptors matching in ORBSLAM3 and SUPERSLAM3.

	FeatureExtraction (ms)	FeatureMatching (ms)
MAX SUPERSLAM3	2.13	41.45
MAX ORBSLAM3	4.35	12.45
AVG SUPERSLAM3	1.75	33.45
AVG ORBSLAM3	3.92	9.23

## Data Availability

Source code will be made available on request.

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
