# Peer review of "Integrating Sparse Learning-Based Feature Detectors into Simultaneous Localization and Mapping—A Benchmark Study"

_sensors, 2023, doi:10.3390/s23042286_

Round 1

Author Response

Dear Reviewer,

Please be informed that we have attached our response letter for your perusal. We hope that this letter adequately addresses any concerns and comments that were raised during the review process.

Thank you for taking the time to provide feedback on our work, which has helped us to improve the quality of our manuscript.

Sincerely,

Giuseppe Mollica, 
Marco Legittimo, 
Alberto Dionigi, 
Gabriele Costante, 
Paolo Valigi.

Author Response

(The authors gave the same response as above.)

Round 2

Reviewer 1 Report

After the review addressed by the authors, I consider this paper suitable to be published.